# In Vitro Effect of the Common Culinary Herb Winter Savory (*Satureja montana*) against the Infamous Food Pathogen *Campylobacter jejuni*

**DOI:** 10.3390/foods9040537

**Published:** 2020-04-24

**Authors:** Katarina Šimunović, Franz Bucar, Anja Klančnik, Francesco Pompei, Antonello Paparella, Sonja Smole Možina

**Affiliations:** 1Department of Food Science and Technology, Biotechnical Faculty, University of Ljubljana, Jamnikarjeva 101, 1000 Ljubljana, Slovenia; katarina.simunovic@bf.uni-lj.si (K.Š.); anja.klancnik@bf.uni-lj.si (A.K.); frpompei@gmail.com (F.P.); 2Institute of Pharmaceutical Sciences, Department of Pharmacognosy, University of Graz, Universitätsplatz 4, 8010 Graz, Austria; franz.bucar@uni-graz.at; 3Department of Food Science, Faculty of Bioscience and Technology for Food, Agriculture and Environment, University of Teramo, via Balzarini 1, 64100 Teramo, Italy; apaparella@unite.it

**Keywords:** *Satureja montana* herb, *Campylobacter jejuni*, essential oil, ethanolic extract, antimicrobial, synergistic activity, efflux pump inhibition, membrane disruption, carvacrol, thymol

## Abstract

The culinary herb *Satureja montana*, known as winter savory, is an ingredient of traditional dishes known in different parts of the world. As an ingredient of foods it has the potential to improve their safety. In this study, the herb’s activity was investigated against *Campylobacter jejuni,* the leading cause of the most prevalent bacterial gastroenteritis worldwide. The ethanolic extract and essential oil of the herb were chemically characterized and six pure compounds—carvacrol, thymol, thymoquinone, *p*-cymene, γ-terpinene, and rosmarinic acid—were chosen for further analysis. The antimicrobial activity of the ethanolic extract (MIC 250 mg/L) was 4-fold higher compared to the essential oil. Carvacrol, thymol and thymoquinone had the strongest antimicrobial effect (MIC 31.25 mg/L) and a strong synergistic activity between carvacrol and thymol was determined (FICi 0.2). Strong inhibitory effect on *C.*
*jejuni* efflux pumps (2-fold inhibition) and disruption of membrane integrity (> 80% disruption) of the herb were determined as modes of action. For resistance against the herb, *C.*
*jejuni* need efflux pumps, although increased resistance against this herb does not co-occur with increased efflux pump activity, as for antibiotics. This study shows the potential of a common culinary herb for the reduction of the food pathogen *C.*
*jejuni* without increasing resistance.

## 1. Introduction

*Campylobacter* spp. are the most commonly reported bacterial cause of gastroenteritis in the European Union, which is mainly due to infection by *Campylobacter jejuni* [1,2]. Campylobacteriosis is a disease that presents itself as watery diarrhea, fever and cramps, and can lead to the development of Guillain-Barre syndrome, a severe neurological condition [2,3]. With the number of confirmed campylobacteriosis cases reaching more than 250,000 per year in the EU, this is a burden for human health as well as national economies [1,4].

The present-day antibiotic resistance of bacteria is a considerable challenge, and *C. jejuni* is no exception. Due to the overuse of antibiotics in veterinary and human medicine, *C. jejuni* resistance levels to ciprofloxacin, nalidixic acid, and tetracycline are high. With more than 50% of isolates from poultry now resistant to at least one antibiotic, the risk of resistant *C. jejuni* spreading through the food chain is also high [5]. *C. jejuni* resistance nodulation division (RND) efflux pumps CmeABC and CmeDEF, and the major facilitator superfamily (MFS) efflux pump CmeGH are great contributors to these resistances. These efflux pumps and their overexpression cause resistance against antibiotics, such as erythromycin and ciprofloxacin, heavy metals, and bile acids, although, even without overexpression they can confer other resistance determinants for greater antibiotic and stress resistance in *C. jejuni* [6,7,8,9]. The inhibition of the major efflux pumps can thus lead to increased *C. jejuni* susceptibility to antimicrobials [6]. The worrying resistance levels in bacteria make it necessary to find alternative solutions for treatment and protection against pathogenic bacteria and efflux pumps are a promising target. More than 200 species can be found among the herbs collectively known as savory (*Satureja*), although summer (*S. hortensis*) and winter savory (*S. montana*) are the best known. Although use of the summer savory species is more widespread, winter savory has been gaining in popularity. The reason is the easier cultivation of *S. montana*, even in colder climates, and its more intense taste. *S. montana* is used in traditional dishes in the Karst region of Slovenia [10], but its use is also documented in different parts of the world [11,12,13]. *S. montana* extracts have shown good anti-oxidative potential and antimicrobial activities against *Staphylococcus aureus*, *Escherichia coli, Bacillus cereus*, *Salmonella enterica*, and *Listeria monocytogenes* [14,15,16], and when added to mortadella cured pork, it was shown to reduce the levels of *Clostridium perfringens* [17]. Similar effects were shown for *L. monocytogenes* in wine-marinated beef [16]. These characteristics make *S. montana* an interesting and promising product to study in the fight against foodborne pathogens. It can easily be added to food and thus serve as a supporting therapeutic for bacterial gastroenteritis.

The aim of this study was to determine the antimicrobial activity of the herb *S. montana* as a crude ethanolic extract and essential oil against the major foodborne pathogen *C. jejuni*. Furthermore, six compounds identified in the herb were investigated in their pure form to determine their potential synergistic effects against *C. jejuni*: carvacrol, thymol, thymoquinone, γ-terpinene, *p*-cymene, and rosmarinic acid. The modes of action of the ethanolic extract, essential oil, and the pure compounds were determined in terms of efflux pump inhibition and membrane disruption. Additionally, the involvement of the *C. jejuni* efflux pumps CmeABC, CmeDEF, and CmeGH in *C. jejuni* resistance against *S. montana* was examined

## 2. Materials and Methods 

### 2.1. Ethanolic Extract and Essential Oil Preparation and Chemical Analysis

*S. montana* was supplied as dried herb by Kottas Pharma (Vienna, Austria). Ethanol was selected as solvent due to its ability to extract medium polarity as well as non-polar compounds and its compatibility with food processing. For extraction, 2 L of 96% denatured ethanol (Roth, Karlsruhe, Germany) was added to 100 g of herbal material, then mixed with a magnetic stirrer at room temperature for 48 h and filtered (Rotilabo pleated paper filters; Roth). The filtrate was evaporated to dryness using a rotary evaporator (Rotavapor, Büchi, Flawil, Switzerland) under vacuum at 40 °C and 175 mbar pressure. The final drying was carried out under a nitrogen flow, affording a yield of 3 g of dried extract from 100 g plant material. This ethanolic extract was analyzed by ultra-high-performance liquid chromatography–photodiode array–electrospray ionization mass spectrometry (UHPLC–PDA–ESI-MS). The UHPLC system (Dionex Ultimate 3000 RS; Thermo Scientific, Waltham, MA, USA) included pump, autosampler, column compartment, and PDA detector. Separation was carried out on an analytical column (Zorbax SB-C18 Rapid Resolution HD; 100 × 2.1 mm, 1.8 μm; Agilent, Santa Clara, CA, USA). The mobile phases were water plus 0.1% formic acid (A) and non-acidified acetonitrile (B), with gradient elution of: 0.0 → 9.0 min, 8% → 45% B; 9.0 → 15.0 min, 45% → 100% B; 15.0 → 15.5 min, 100% → 8% B; 15.5 → 21 min, 8% B. The column temperature was 35 °C and the flow rate was 0.390 mL/min. A solution of 5 mg/mL ethanolic extract was prepared in methanol. Control samples of 0.5 mg/mL rosmarinic acid and 0.5 µL/mL carvacrol were used. The injection volume was 2.0 µL. The PDA detection was in the wavelength range of 190 nm to 500 nm. 

MS detection was achieved using a linear ion-trap mass spectrometer (LTQ XL; Thermo Scientific) equipped with an ESI source (Thermo Scientific). Mass spectra were recorded in negative and positive ion modes for the m/z range from 50 amu to 2000 amu, with data-dependent fragmentation (normalized collision energy, 35%). The further settings included: source voltage, 3.5 kV (ESI negative), 5.0 kV (ESI positive); capillary temperature, 350 °C; source temperature, 300 °C; sheath gas flow, 40 units (machine setting); auxiliary gas flow, 10 units (machine setting). 

The essential oil was collected after hydro-distillation (Clevenger apparatus) of 30 g of plant material for 2 h. Gas chromatography (GC)-MS analysis was performed using a GC system (7890A; Agilent) interfaced to a mass selective detector (5975C; Agilent), with: electron ionization, 70 eV; ion source temperature, 230 °C; and interface temperature, 280 °C. The split injection (injection volume, 0.2 µL; split ratio, 50:1) at 240 °C injected the sample onto a fused silica capillary column (5% phenyl, 95% methyl polysiloxane; HP-5MS; 30 m × 250 µm × 0.25 µm; Agilent). Quantification of the compounds of interest was performed by comparison of the relative peak areas after normalization obtained by GC–flame ionization detector (FID) analysis, using the area % method without considering calibration factors. A gas chromatograph (GC 2014; Shimadzu, Kyoto, Japan) equipped with a FID and a fused silica column (5% polysilarylene, 95% polymethylsiloxane; Zebron ZB-5MSi; 30 m × 250 µm × 0.25 µm; Phenomenex, Torrance, CA, USA) was used, with experimental conditions identical to those for the GC-MS analysis. Reference compounds were obtained from Sigma Aldrich (Munich, Germany) with the following purities: carvacrol 98.0%, thymol ≥ 98.5%, thymoquinone ≥ 98.0%, γ-terpinene 98.0%, *p*-cymene 98.0%, and rosmarinic acid ≥ 98.0%.

### 2.2. Bacterial Strains and Growth Conditions

The *C. jejuni* strains shown in Appendix A were stored at −80 °C in 20% glycerol and 80% Mueller Hinton broth (MHB; Oxoid, Basingstoke, UK) and were grown on Mueller-Hinton agar (MHA; Oxoid) at 42 °C under microaerobic conditions (5% O_2_, 10% CO_2_, in N_2_) for 24 h. The second passage of each culture was used in the experiments. When necessary, MHA was supplemented with 30 mg/L kanamycin (Sigma Aldrich) or 4 mg/L chloramphenicol (Merck, Darmstadt, Germany).

### 2.3. Preparation of the C. jejuni 11168ΔcmeG Mutant Strain

Natural transformation of *C. jejuni* [18] was used to generate the *cmeG* insertional mutant strain. The donor DNA was *C. jejuni* genomic DNA prepared from a previously published mutant strain [6], and the recipient strain was *C. jejuni* 11168. The *cmeG* transformants were selected on MHA with 30 mg/L kanamycin, and their DNA was isolated following the protocol described by Klančnik et al. [19]. Successful transformation was confirmed by PCR using the specific primers Cj1375F (CATCTACACAAATGCCACTATCATCACTTAA) and Cj1375R (GCCACAAGCCAAATTAGAGC TAAAATTACTAA), as described previously [6].

### 2.4. Antimicrobial Activity Assay

The minimal inhibitory concentrations (MICs) of *S. montana* ethanolic extract and essential oil, and of the pure compounds carvacrol, thymol, thymoquinone, γ-terpinene, *p*-cymene and rosmarinic acid were determined using the broth microdilution method as described previously [6] for *C. jejuni* 11,168 and its efflux pump mutants *∆cmeB, ∆cmeR, ∆cmeF,* and *∆cmeG.* All tested compounds were freshly dissolved in 100% dimethylsulfoxide (DMSO, Sigma Aldrich) before the test. For broth microdilution the tested compounds prepared in DMSO were diluted in MHB (to a final concentration of 2.5% DMSO). The bacterial respiration was determined with 10 μL of resazurin solution (0.25 mg/mL resazurin, 0.14 mg/mL menadion, in MHB; Sigma Aldrich). As resazurin detects *C. jejuni* numbers over 5 × 10^5^ CFU/mL, the MICs were assessed as the lowest concentration of a compound at which bacterial growth/respiration was inhibited. All tests were carried out in triplicate.

### 2.5. Checkerboard Assay

The checkerboard method was used to determine the potential synergistic activity of the pure compounds carvacrol, thymol, thymoquinone, γ-terpinene, *p*-cymene, and rosmarinic acid on *C. jejuni* 11168, as described previously [20]. Briefly, the tested compounds were prepared in MHB with 2.5% DMSO, and serially diluted in a microtiter plate, where the first compound was two-fold serially diluted in columns, and the second in rows, to a final volume of 100 µL. Then 100 µL *C. jejuni* 11,168 culture in MHB at 5 × 10^5^ CFU/mL was added to each well. The plates were incubated at 42 °C under microaerobic conditions for 24 h. The MICs of the combinations of compounds were determined with the resazurin solution as described for the antimicrobial activity assay. The tests were carried out in triplicate. The fractional inhibitory concentration index (FIC_I_) was used as the measure of the synergistic activity of these compounds. The FIC_I_ is defined as the sum of FIC_A_ and FIC_B_ as it is shown in Equation (3), where FIC_A_ is the MIC of compound A supplemented with compound B (A_mix_), divided by the MIC of compound A alone (A_pure_), as shown in Equation (1), and FIC_B_ the MIC of compound B supplemented with compound A (B_mix_), divided by the MIC of compound B alone (B_pure_), as shown in Equation (2):FIC_A_ = MIC_Amix_/MIC_Apure_(1)
FIC_B_ = MIC_Bmix_/MIC_Bpure_(2)
FIC_I_ = FIC_A_ + FIC_B_(3)

These values were interpreted as follows: for FIC_I_ ≤ 0.5, a synergistic effect; for FIC_I_ > 0.5 and ≤ 4, an additive effect; and for FIC_I_ ˃ 4, an antagonistic effect [21].

### 2.6. Ethidium Bromide Accumulation Assay

To determine whether *S. montana* ethanolic extract and essential oil, and the pure compounds were efflux pump inhibitors against *C. jejuni* 11168, an ethidium bromide accumulation assay was carried out, as previously described by Kovač et al. [22], with some modifications. Briefly, cultures of *C. jejuni* 11,168 previously adjusted to OD_600_ 0.2 in MHB and incubated at 42 °C under microaerobic conditions for 4 h were washed and resuspended in phosphate-buffered saline (Oxoid), and adjusted to OD_600_ 0.4. The prepared ethanolic extract, essential oil, and pure compounds were added to the cultures to a final volume of 100 µL in black microtiter plates (Nunc, Thermo Scientific). Then 0.5 mg/L ethidium bromide (final concentration; Sigma Aldrich) was added. Ethidium bromide is an intercalating agent that binds to the DNA strand within the bacterial cell and emits fluorescence. The accumulation of intracellular ethidium bromide was measured as increasing fluorescence at ʎ_ex_ 500 nm and ʎ_em_ 608 nm, using a microplate reader (Varioskan Lux; Thermo Fisher Scientific) at 1 min intervals over 1 h. As the positive control, the reference efflux pump inhibitor reserpine (Sigma Aldrich) at 10 mg/L was used. The fold-changes in the ethidium bromide accumulation determined after 1 h were used as the measure of the efflux pump inhibition. These measurements were carried out in triplicate. 

### 2.7. Membrane Integrity Assay

The disruptive effects of *S. montana* ethanolic extract and essential oil, and the pure compounds (i.e., carvacrol, thymol, thymoquinone, γ-terpinene, *p*-cymene, rosmarinic acid) on the membranes of *C. jejuni* 11,168 were evaluated using Live/Dead bacterial viability kits (L-7012; Molecular Probes, Eugene, OR, USA), as previously described [22]. Briefly, mid-exponential phase cells (i.e., after an 8-h incubation) were harvested and washed twice with warm phosphate-buffered saline (42 °C), with OD_600_ adjusted to 0.4. The compounds were added to the prepared cultures at sub-inhibitory concentrations, with no effects on *C. jejuni* growth. The Live/Dead solution was prepared according to the manufacturer instructions and added to the cultures (1:1, *v*/*v*). The SYTO9 fluorescence was measured at λ_ex_ 481 nm and λ_em_ 510 nm at 60 s intervals over 1 h, in black microtiter plates (Nunc). A heat-treated culture (80 °C, 15 min) was used as the negative control, and an untreated culture as the positive control. The relative rates of disruption were calculated considering 100% disruption in the heat-treated culture and 0% disruption in the untreated culture. These experiments were carried out in triplicate.

### 2.8. Statistics

Statistical analysis was carried out using the SPSS software version 22 (IBM, Armonk, NY, USA). Ethidium bromide accumulation and membrane disruption were compared between untreated and treated cultures using Student’s *t*-tests or Mann Whitney U tests. In the checkerboard assays, significant synergism was set at FIC_I_ ≤ 0.5. Differences in the MICs of the tested compounds between *C. jejuni* 11,168 wild type and the mutant strains at ≥2 were set as biologically significant.

## 3. Results

### 3.1. Chemical Analysis

The *S. montana* ethanolic extract was analysed by UHPLC with PDA and ESI-MS detection. Appendix A shows the peak assignments with UV and MS data, and Figure 1 shows a total scan UV chromatogram. Rosmarinic acid (Figure 1, peak 1) was identified as a major constituent of the ethanolic extract. Carvacrol (Figure 1, peak 7) and thymoquinone (Figure 1, peak 6) followed next in terms of quantities. In addition, a caffeoyl rosmarinic acid isomer was identified. A number of methoxylated flavones were identified according to their UV data and their fragmentation patterns with ESI-MS (positive and negative modes); however, to determine the definite substitution patterns, isolation of the pure compounds and structural elucidation by nuclear magnetic resonance would be necessary. Although no easily volatile compounds were detected in the ethanolic extract it became apparent that less volatile compounds of the essential oil like carvacrol and thymoquinone were still present. Hence, in addition the composition of the essential oil was analyzed by GC-MS and GC-FID (Appendix A). According to these analyses, carvacrol, thymol, thymoquinone, *p*-cymene, γ-terpinene, and rosmarinic acid were selected as the individual and combined pure test compounds.

### 3.2. Anti-Campylobacter Activity of S. montana Ethanolic Extract and Essential Oil

The antimicrobial activity of the *S. montana* ethanolic extract and essential oil and the six chosen pure compounds were tested against *C. jejuni* 11,168 (Table 1). The ethanolic extract and essential oil had medium anti-*Campylobacter* activity, with MICs of 250 mg/L and 1000 mg/L, respectively.

Further, the pure carvacrol, thymol, and thymoquinone had greater antimicrobial activities (MICs of 31.25 mg/L for all three) compared to the ethanolic extract and essential oil. The opposite was seen for the pure *p*-cymene and γ-terpinene (MICs of 1000 mg/L for both), which showed 4-fold greater MICs compared to the ethanolic extract, but no difference compared to the essential oil. Rosmarinic acid (MIC 250 mg/L) was as active as the ethanolic extract, with a lower MIC compared to the essential oil.

### 3.3. Synergistic Activity of the Six Pure Compounds

The potential synergistic activity of carvacrol, thymol, thymoquinone, *p*-cymene, γ-terpinene, and rosmarinic acid were examined using the checkerboard assay and determined through the calculation of the FIC_I_ values as: FIC_I_ ≤ 0.5, synergistic; FIC_I_ > 0.5 and ≤ 4, additive; and FIC_I_ ˃ 4, antagonistic [21].

The most effective combinations of these pure compounds that showed strong synergistic activity according to the FIC_I_ were (see Table 2 for details): carvacrol plus thymoquinone (FIC_I_, 0.5), carvacrol plus thymol (FIC_I_, 0.2), thymol plus thymoquinone (FIC_I_, 0.3), and thymoquinone plus rosmarinic acid (FIC_I_, 0.5). A second combination of carvacrol (3.9 µg/mL) plus thymol (7.81 µg/mL) was also synergistic (FIC_I_, 0.3). All the other tested combinations showed additive effects, with no antagonistic effects seen. The antimicrobial activity of carvacrol, thymoquinone, and rosmarinic acid in these synergistic combinations was significantly increased compared to the pure compounds alone.

### 3.4. Efflux Pump Inhibition in C. jejuni

The efflux pump inhibitory effects of *S. montana* ethanolic extract and essential oil, and their six major constituents, were determined in *C. jejuni* 11,168, using the ethidium bromide accumulation assay. Ethidium bromide accumulation is used as a measure of efflux pump activity of a culture. When ethidium bromide accumulates in the cells it is toxic, and therefore the cells use their efflux pumps to eliminate it. Within the cell, the intercalation of ethidium bromide in the DNA strand results in a fluorescent signal. Thus, for this assay, higher fluorescence indicates higher accumulation of ethidium bromide, which in turn translates into lower efflux pump activity, and hence their inhibition.

The highest tested concentration of *S. montana* ethanolic extract (125 mg/L) resulted in a 1.9-fold increase in *C. jejuni* ethidium bromide accumulation after 1 h of treatment (Figure 2A). The greatest accumulation of ethidium bromide, 2.07-fold as compared to the untreated control (Figure 2B), was observed for the cultures treated with *S. montana* essential oil at 0.5× MIC (500 mg/L). Both the ethanolic extract and the essential oil showed a stronger effect than reserpine (1.39-fold), the known efflux pump inhibitor, and in both cases the effects were concentration dependent, as at lower concentrations they had lesser effects on ethidium bromide accumulation.

The treatment of *C. jejuni* with the ethanolic extract and essential oil at 0.25× MICs increased the ethidium bromide accumulation by 1.57-fold and 1.79-fold, respectively (Figure 2C). Across all six pure compounds, only *p*-cymene showed any appreciable efflux pump inhibitory activity, with a 1.38-fold increase in ethidium bromide accumulation. Carvacrol had a weak effect, with a 1.26-fold increase in ethidium bromide accumulation. Thymol, thymoquinone, γ-terpinene, and rosmarinic acid only increased the ethidium bromide accumulation by up to 1.2-fold.

### 3.5. Disruption of Membrane Integrity in C. jejuni

All the tested concentrations of *S. montana* ethanolic extract and essential oil showed disruptive effects on the membranes of *C. jejuni* 11,168, and these effects were concentration dependent. At 0.5× MICs of *S. montana* ethanolic extract and essential oil, treatment for 1 h decreased *C. jejuni* membrane integrity by 88% and 83%, respectively (Figure 3A,B). The concentrations of 0.25× MICs for all these samples were chosen for further testing, as they had no effects on the growth of *C. jejuni*. The ethanolic extract and essential oil at 0.25× MICs damaged the *C. jejuni* membranes by 49% and 64%, respectively. Of the tested pure compounds, again at 0.25× MICs, only rosmarinic acid showed any disruption of the membranes (15%). 

### 3.6. Involvement of Efflux Systems in Resistance of C. jejuni to S. montana Ethanolic Extract and Essential Oil and Their Six Major Constituents

To examine the involvement of the efflux systems of *C. jejuni* in its resistance to the antimicrobial activity of *S. montana* ethanolic extract and essential oil, and their six major constituents, their MICs were determined for wild-type *C. jejuni* 11,168 and for the efflux pump mutants: Δ*cmeF*, which lacks a functional CmeDEF efflux pump; Δ*cmeG*, which lacks a functional CmeGH efflux pump; Δ*cmeB*, which lacks a functional CmeABC efflux pump; and Δ*cmeR*, which lacks a functional CmeR repressor of the CmeABC efflux pump (Table 3).

Here, the CmeGH efflux pump had a major role in *C. jejuni* resistance to both *S. montana* ethanolic extract and essential oil, whereas the CmeABC efflux pump was involved only in resistance to the ethanolic extract. The antimicrobial activity of *S. montana* ethanolic extract against *C. jejuni* Δ*cmeB* and Δ*cmeG* was 4-fold greater than for *C. jejuni* wild-type. The lack of the CmeDEF efflux pump resulted in only a halving of the MIC of the ethanolic extract, with no changes seen for the activity of the essential oil.

Inactivation of the CmeABC efflux pump resulted in 4-fold the antimicrobial activity of both thymol and rosmarinic acid, and in 2-fold the antimicrobial activity of both carvacrol and γ-terpinene. Inactivation of the CmeDEF efflux pump increased the γ-terpinene antimicrobial activity by 4-fold, and to a lesser degree (2-fold) for carvacrol, thymoquinone, *p*-cymene, and rosmarinic acid. The highest impact on the *C. jejuni* resistance to the pure compounds was for the inactivation of the CmeGH efflux pump. Here, carvacrol and thymoquinone showed 32-fold increased antimicrobial activities in the mutant strain compared to the wild-type, with 16-fold increase seen for thymol. This indicates the strong involvement of the CmeGH efflux pump in the *C. jejuni* resistance to these compounds. However, the effects of inactivation of the CmeGH efflux pump on the antimicrobial activity of *p*-cymene, γ-terpinene, and rosmarinic acid were minor (2-fold increase). Surprisingly, the CmeR deficient mutant was more sensitive to carvacrol, thymol, thymoquinone, γ-terpinene, and rosmarinic acid, with 4-fold antimicrobial activities seen.

## 4. Discussion

In this study, we analyzed the anti-*Campylobacter* activity of *S. montana* with a comprehensive approach in which we chemically characterized the ethanolic extract and essential oil and their antimicrobial action against *C. jejuni.* The modes of action of *S. montana* and its major constituents were determined, but also the importance of efflux pump involvement in *Campylobacter* resistance against *S. montana* was shown.

The chemical composition and the anti-*Campylobacter* activity of an ethanolic extract and an essential oil from the common culinary herb winter savory (*S. montana*) were examined. Rosmarinic acid was the major compound in the ethanolic extract, already reported as one of the most common caffeic acid esters in *Lamiaceae* [23]. In addition, a caffeoyl rosmarinic acid isomer was detected. It’s UV and MS data are similar to those for melitric acid A, which was recently described for *Satureja biflora* [24]. Carvacrol was seen to make up about 63% of the essential oil, as already observed by other authors [25], and was also present as a major compound in the ethanolic extract, together with thymoquinone. Conversely, thymoquinone was found at a low level in the essential oil (1.8%); however, it was extracted as a higher proportion of the ethanolic extract and might be of relevance for anti-*Campylobacter* effects.

*S. montana* ethanolic extract and essential oil proved to have good anti-microbial activity against *C. jejuni*, higher than what was reported on other foodborne pathogens such as *E. coli*, *S. aureus*, and *B. cereus* [15]. These differences might be due to the different origin of the same plant, the different strain investigated, and/or the reaction of these foodborne pathogens to the extracts. The difference between antimicrobial activity of the ethanolic extract and essential oil (4-fold) against *C. jejuni* may be due to their different compositions and thus differences in their stability in an aqueous solution with an organic solvent, as they were prepared in this study.

*S. montana* ethanolic extract and essential oil were seen to contain many components, and based on the chemical analyses, the pure forms of the six major components were investigated individually, as: carvacrol, thymol, thymoquinone, *p*-cymene, γ-terpinene, and rosmarinic acid. The antimicrobial and synergistic activity of these compounds ranged from very strong to weak. In detail, for carvacrol and thymol, whose anti-*Campylobacter* activity was also studied by Alphen et al. [26], Kelly et al. [27], Upadhyay [28] and others, our data confirmed the published data, in terms of good antimicrobial activity against *C. jejuni*. On the other side, when thymoquinone was studied for its activity against *S. aureus* and *P. aeruginosa*, no effects were observed, although it was reported to be effective against *Vibrio parahaemolyticus* [29] and *E. coli* [30]. In the present study, thymoquinone showed antimicrobial activity against *C. jejuni* that was comparable to carvacrol and thymol, which also confirms the better antimicrobial activity of thymoquinone against Gram-negative bacteria. *p*-cymene has been shown to be effective against the enteric pathogens *L. monocytogenes*, *E. coli*, and *Salmonella* spp. [31], but its anti-*Campylobacte*r activity in the present study was weak. Weak antimicrobial activity of γ-terpinene against *E. coli* and *S. aureus* was reported by Burt et al. [32], which is confirmed in the present study for *C. jejuni*. Finally, rosmarinic acid, a well-known polyphenol, has been shown to have antibacterial activity against several foodborne pathogens as *S. aureus*, *E. coli*, and *Shigella* spp. [33]. The results of this study confirmed the observations of Klančnik et al. [19] of a medium anti-*Campylobacter* activity of rosmarinic acid.

The major constituents of *S. montana* ethanolic extract and essential oil had both strong and weak antimicrobial activity, whereby the overall activity of *S. montana* ethanolic extract and essential oil reflects the antimicrobial activities of the pure compounds.

Natural extract and essential oils are mixtures of many different compounds, and thus synergistic or antagonistic activities of their constituents are likely. Previous studies have shown increased activity for *p*-cymene when combined with carvacrol [34], and for carvacrol in combination with thymol [32,35], which were confirmed for *C. jejuni* in the present study. Here, synergism was also seen for the combinations of thymoquinone with carvacrol, thymol, and rosmarinic acid. These tested major constituents of *S. montana* have been shown to affect the cell membrane of bacteria [36]. Here we see that, even when we combine two compounds with apparently similar action, e.g., carvacrol and thymol, their antimicrobial activity will be potentiated. This points to different targets of these constituents in the *C. jejuni* cell. 

This study shows that the antimicrobial activity of the *S. montana* ethanolic extract and essential oil cannot be attributed to the action of any one single compound but will be the result of the combined, be it strong or weak, effects of their constituents. It should be noted that only binary combinations of only six pure compounds were tested here, therefore we cannot conclude what the exact effect of multiple combinations in one extract/essential oil would be. In *C. jejuni*, efflux pump inhibitors can increase its sensitivity to various antimicrobials, including antibiotics [22,37]. The inhibitory effects on the *C. jejuni* efflux pumps of *S. montana* ethanolic extract were weak compared to those of the resistance modulator α-pinene [22], which is also a constituent of *S. montana* herb, but were still stronger than those of the known efflux pump inhibitor reserpine. This makes *S. montana* ethanolic extract an important subject for future consideration in studies on resistance modulation of *C. jejuni*. 

The inhibitory effects on *C. jejuni* efflux pumps of *S. montana* ethanolic extract and essential oil were not reflected in the activity of the six pure compounds tested in this study (Figure 2). This is an important issue to consider when searching for new efflux pump inhibitors derived from natural extracts, as the pure compounds might not reflect the activity of the whole phytocomplex.

The Lamiaceae family has been known to harbor plants with vast biological properties, including antimicrobial activity. Modes of action of these essential oils and their constituents are targeting the cell membrane with protein degradation and membrane disruption [36]. The Gram-negative cell membrane is a formidable barrier for antimicrobials to bypass, and thus the search for membrane disruptors is an important challenge. Both *S. montana* ethanolic extract and essential oil had membrane disruptive effects on *C. jejuni*, although again, these were not confirmed by data of the six pure compounds (Figure 3). Indeed, although La Storia et al. [38] reported that carvacrol acts as a membrane disruptor, the present study does not confirm. In this respect, it should be taken into account that we tested the modes of action at sub-inhibitory concentrations, in contrast to the higher inhibitory concentrations that are usually evaluated.

In particular, the efflux pump inhibitory effects and membrane disruptive effects of *S. montana* ethanolic extract and essential oil, and the lack of these effects in the pure compounds tested, indicate the presence of further compounds in the ethanolic extract that are present at lower concentrations, but that act as potentially more potent efflux pump inhibitors and membrane disruptors.

When a new antimicrobial compound is tested, the resistance mechanisms of bacteria against the compound in question need to be considered. It has been shown that *C. jejuni* major efflux pumps that are involved in resistance against antibiotics and bile salts [6,8] also contribute to the resistance of *C. jejuni* to natural extracts and phenolic compounds [19,22]. In the present study, both the CmeABC and CmeDEF efflux pump systems were seen to be involved in the resistance against *S. montana* ethanolic extract and essential oil, and their major constituents, which is not surprising, as their substrates often overlap [39]. Interestingly, the *C. jejuni* 11168∆*cmeR* mutant showed increased sensitivity to the tested compounds. CmeR acts as a repressor of the promotor of the CmeABC multidrug efflux pump, and mutation of the *cmeR* gene leads to enhanced production of this pump, and higher resistance of *C. jejuni* to antibiotics [8,40]. This shows the involvement of the CmeABC efflux pump in the excretion of these compounds, and its importance as part of the *C. jejuni* resistance to *S. montana* ethanolic extract and essential oil*,* although there is no threat for the development of a higher resistance against *S. montana* with increased efflux.

The involvement of the CmeGH efflux pumps in the resistance of *C. jejuni* to *S. montana* ethanolic extract and essential oil is similar to that of the CmeABC and CmeDEF efflux pumps. Interestingly, the involvement of the CmeGH efflux pump was most prominent in the resistance against thymol, carvacrol, and thymoquinone. This increase in sensitivity of the mutant strain that lacked a functional CmeGH efflux pump was much higher than previously reported for antibiotic susceptibility of *C.jejuni* 11,168∆*cmeG* mutant strain. Jeon et al. [6] showed increased susceptibility of the *C. jejuni* 11168∆*cmeG* mutant strain to erythromycin (8-fold), ciprofloxacin (4-fold), gentamycin (16-fold), tetracycline (4-fold), and ethidium bromide (8-fold). In the present study, there was a 32-fold increase in the susceptibility of *C. jejuni* 11168∆*cmeG* mutant strain for carvacrol and thymoquinone. Our data show, for the first time, a major role for the CmeGH efflux pump in the resistance of *C. jejuni* against carvacrol, thymol, and thymoquinone, and therefore against natural extracts and essential oils that contain these compounds.

## 5. Conclusions

Although the antimicrobial activity of the pure compounds surpassed that of either *S. montana* ethanolic extract and essential oil, this was not reflected in the efflux pump inhibition or the membrane disruption. This shows how such a complex mixture of different compounds cannot be described only by its major constituents, and that the action of multiple compounds is necessary to achieve the desired effects. With their good efflux pump inhibitory activity, *S. montana* ethanolic extract and essential oil show promise as resistance modulators. Although to confirm this, further testing is necessary, including the possible synergistic activity of the extracts with antibiotics of clinical importance (e.g., ciprofloxacin and erythromycin) in vitro and *in vivo*. The *C. jejuni* major efflux pump also has a vital role in *C. jejuni* resistance to *S. montana* ethanolic extract and essential oil, although the present study does not allow us to conclude that the occurrence of resistance against *S. montana* ethanolic extract and essential oil would be due to increased CmeABC efflux pump activity. However, this study demonstrates the use of natural compounds to increase *Campylobacter* susceptibility, and thus the possibility of alternative control strategies.

## Figures and Tables

**Figure 1 foods-09-00537-f001:**
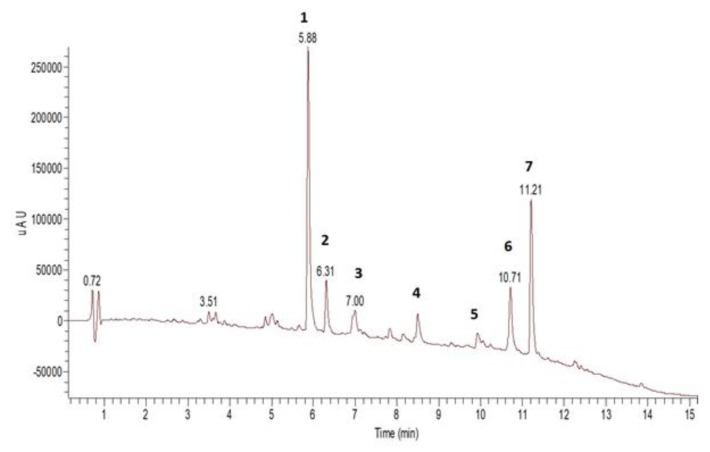
UHPLC separation of *Satureja montana* ethanolic extract. Total scan UV chromatogram (190-500 nm). Peak assignments: 1 = rosmarinic acid, 2 = caffeoyl rosmarinic acid isomer, 3 = apigeninmethyl ether hexosyldeoxyhexoside, 4 = trihydroxy-trimethoxyflavone, 5 = dihydroxy-dimethoxyflavone, 6 = thymoquinone, 7 = carvacrol.

**Figure 2 foods-09-00537-f002:**
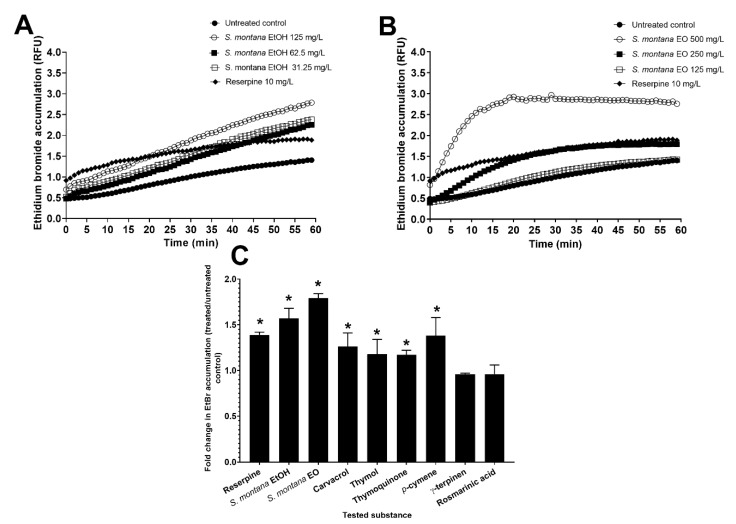
Effect of *S. montana* (**A**) ethanolic extract (EtOH) and (**B**) essential oil on membrane integrity disruption in *C. jejuni* NCTC 11,168 in sub-inhibitory concentrations of 0.5× (125 and 500 mg/L), 0.25× (62.5 and 250 mg/L), and 0.125× (31.25 and 125 mg/L) of the minimal inhibitory concentration (MIC), presented in relative fluorescent units (RFU) measured for 1 h in 1 min intervals. Fold increase in EtBr accumulation (**C**) in the treated culture compared to untreated culture is shown for *S. montana* ethanolic extract and essential oil, and 6 pure compounds, at the subinhibitory concentration of 0.25× MIC. Reserpine at 10 mg/L was used as a positive control. A higher value presents higher accumulation of ethidium bromide and thus a lower activity and inhibition of *C. jejuni* efflux pumps. *, *p* < 0.05 (Student’s *t*-test).

**Figure 3 foods-09-00537-f003:**
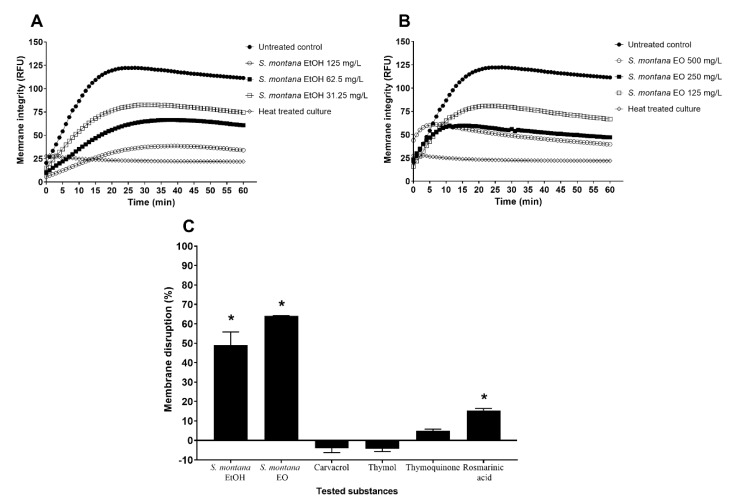
Effect of *S. montana* (**A**) ethanolic extract (EtOH) and (**B**) essential oil (EO) on membrane integrity disruption of *C. jejuni* 11,168 in sub-inhibitory concentrations of 0.5 × (125 and 500 mg/L), 0.25 × (62.5 and 250 mg/L), and 0.125 × (31.25 and 125 mg/L) of the minimal inhibitory concentration (MIC), presented in relative fluorescent units (RFU) measured for 1 h in 1 min intervals. A higher value presents higher accumulation of ethidium bromide and thus a lower activity and inhibition of *C. jejuni* efflux pumps. The disruption of *C. jejuni* membrane (**C**) in the treated culture compared to the untreated culture, is shown for *S. montana* ethanolic extract and essential oil, and 6 pure compounds, at the subinhibitory concentration of 0.25× MIC as the percentage (%) of disruption. Membrane disruption of *p*-cymene and *γ*-terpinene could not be measured because of autofluorescence of compounds. *, *p* < 0.05 (Student’s *t*-test).

**Table 1 foods-09-00537-t001:** Antimicrobial activity of *Satureja montana* ethanolic extract and essential oil, and of the six pure compounds against *Campylobacter jejuni* 11168, in terms of their minimal inhibitory concentrations (MICs) in mg/L.

Treatment	MIC (mg/L)
Ethanolic extract	250
Essential oil	1000
Carvacrol	31.25
Thymol	31.25
Thymoquinone	31.25
*p*-cymene	1000
*γ*-terpinene	1000
Rosmarinic acid	250

**Table 2 foods-09-00537-t002:** Analysis of the synergistic activity of the six pure compounds tested as binary combinations (compound A + compound B) against *Campylobacter jejuni* 11168, in terms of the fractional inhibitory concentration index (FIC_I_) calculated from the minimal inhibitory concentrations (MICs) of the pure compounds alone (A_pure_, B_pure_) and when tested together (A_mix_, B_mix_). See Equations (1)–(3) for further details.

Combination		MIC (mg/L)				FICi
		Pure Compounds		Mixed Compounds		
Compound A	Compound B	A_pure_^a^	B_pure_^b^	A_mix_^c^	B_mix_^d^	
Carvacrol	Thymol	62.5	31.25	7.81	1.95	0.2
	Thymoquinone	62.5	31.25	15.62	7.81	0.5
	*p-*cymene	62.5	500	62.5	500	2.0
	*γ-*terpinene	31.25	2000	31.25	2000	2.0
	Rosmarinic acid	31.25	250	31.25	250	2.0
Thymol	Thymoquinone	31.25	31.25	3.9	3.9	0.3
	*p*-cymene	31.25	31.25	31.25	31.25	2
	γ*-*terpinene	125	1000	62.5	1000	1.5
	Rosmarinic acid	31.25	250	31.25	250	2
Thymoquinone	*p*-cymene	15.62	1000	7.81	1000	1.5
	γ*-*terpinene	31.25	1000	7.81	500	0.8
	Rosmarinic acid	31.25	250	7.81	62.50	0.5
*p*-cymene	γ*-*terpinene	250	1000	125	1000	1.5
	Rosmarinic acid	1000	250	500	125	1.0
γ*-*terpinene	Rosmarinic acid	1000	250	1000	125	1.5

**^a^**—MIC of compound A when tested alone (e.g., carvacrol alone); **^b^**—MIC of compound B when tested alone (e.g., thymol); **^c^**—the concentration of compound A in AB mixture (e.g., carvacrol in carvacrol + thymol mixture) at MIC of AB; **^d^**—the concentration of compound B in AB mixture (e.g., thymol in carvacrol + thymol mixture) at MIC of AB.

**Table 3 foods-09-00537-t003:** Antimicrobial activity of *Satureja montana* ethanolic extract and essential oil, and of the six pure compounds against the *Campylobacter jejuni* 11,168 efflux pump mutants, in terms of their minimal inhibitory concentrations (MICs). Δ*cmeB*, lacking functional efflux pump CmeABC; Δ*cmeR*, lacking repressor of efflux pump CmeABC; Δ*cmeF*, lacking functional efflux pump CmeDEF; and Δ*cmeG*, lacking functional efflux pump CmeGH.

Treatment	Δ*cmeB*		Δ*cmeR*		Δ*cmeF*		Δ*cmeG*	
	MIC (mg/L)	FC^a^	MIC (mg/L)	FC^a^	MIC (mg/L)	FC^a^	MIC (mg/L)	FC^a^
Ethanolic extract	31.25	4	125	1	62.5	2	31.25	4
Essential oil	1000	1	1000	1	1000	1	500	2
Carvacrol	15.25	2	7.81	4	15.62	2	0.97	32
Thymol	7.81	4	7.81	4	31.25	1	1.95	16
Thymoquinone	15.25	2	7.81	4	15.62	2	0.97	32
*p*-cymene	1000	1	500	2	500	2	500	2
*γ-*terpinene	500	2	250	4	250	4	500	2
Rosmarinic acid	62.5	4	125	4	125	2	125	2

**^a^**FC, Fold change increase in antimicrobial activity of the tested compound in the mutant strain, compared to wild type.

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
