# Peer review of "In Vitro Effect of the Common Culinary Herb Winter Savory (Satureja montana) against the Infamous Food Pathogen Campylobacter jejuni"

_foods, 2020, doi:10.3390/foods9040537_

Round 1

Reviewer 1 Report

I suggest to the authors to amend the title of the manuscript

In vitro effect of Satureja montana against the food pathogen Campylobacter jejuni

Abstract line 23-24

Campylobacter jejuni, the most prevalent cause of bacterial gastroenteritis worldwide. 

This is a statement that suits better to the Introduction providing the a supporting current reference.

Author Response

Response to Reviewer 1 Comments

Point 1: I suggest to the authors to amend the title of the manuscript

In vitro effect of Satureja montana against the food pathogen Campylobacter jejuni

Response 1: Thank you for the suggestion. The authors have decided to somewhat modify the title to: In vitro effect of the common culinary herb winter savory (Satureja montana) against the infamous food pathogen Campylobacter jejuni.

Point 2: Abstract line 23-24

Campylobacter jejuni, the most prevalent cause of bacterial gastroenteritis worldwide. 

This is a statement that suits better to the Introduction providing the a supporting current reference.

Response 2: The sentence has been adjusted for clarification, to “…Campylobacter jejuni, the leading cause of the most prevalent bacterial gastroenteritis worldwide.” L24-25.

Reviewer 2 Report

== MAIN ==

- Assay type and interpretation 1.

l.27, tables 1 and 3, text
MIC of the ethanolic extract is 4x 'better' than of essential oil.
-> It's possible that non-volatile phenolics are effective inhibitors of different enzymes (including also efflux pumps)
but
-> How to explain the high values of MIC for EO?
-> Can you consider insolubility/instability of solution/suspension during the assay? E.g., lack of stable mixing with assay broth? Evaporation of essential oil? Is the broth microdilution method in its unmodified version really applicable to essential oils? Can we compare soluble and non-soluble cpds with the same assay? Those are well-known problems.
-> Refer to the literature of this area. Discuss.
similarly
We know that p-cymene and g-terpinene posses other water-solubility than phenols. The situation can be, in part, similar to essential oil.
-> Please, refer to this problem in Discussion.

- Assay type and interpretation 2.

The essential oil has a much lower value of MIC (table 1, table 3).
-> Why, if its constituents work at least not-antagonistic (as a result of the other experiment, table 2)? Carvacrol and g-terpinene are 63+19 R% of essential oil.
-> Discussion and interpretation needed.

- State of the art in Intro

-> What do we already know about the analyzed cpds AND closely related plants (Lamiaceae, THQ, THL, CRL, gTP, pCY, RA) in the aspect of actions reported in this work?

== OTHER ==

Intro, l.74
The journal is devoted to 'foods.'
-> The hermetic abbreviations should be explained here or at the end list of abbrev.
-> Efflux pump types should be briefly described in Intro.

l.78
-> The reason to use 96% ethanol as extrahent should be explained.
-> This procedure does not guarantee high oil content; was the level of oil determined in the extract?

l.88
-> To be clear, explain if the acetonitrile was also acidified.

fig1 caption, tabS2
-> 'Isomer' is unnecessary in all cases.
-> If no standards for ms2 available, how can you know that it is rhamnoside but hexosyl. Shouldn't you name it deoxyhexoside for provisional identification, if you don't have a standard?
-> Correct.

Materials and Methods
-> The purity of standards must be given (with briefly mentioned assay, e.g., GC-MS [producer declaration / internal protocol(reference)], HPLC-UV (program, lambda=...nm) or qNMR (standard, solvent, analytic shift position)).
-> Preparation and stability of stocks of water-not-well-soluble substances (THQ, THM, CRL...) should be given.

== EDITORIAL ==

l.26, 312, elsewhere
'y-terpinene' use special character 'gamma' instead 'y'; remember about italics

l.131, 139, elsewhere
'ρ-'cymene -> 'p-'cymene
use italics for 'p-', everywhere

decide to regularly use or not to use caps after a hyphen, in chemical names

use non-breakable space between C. and jejuni, everywhere

l.261
last line of tab2, correct 'terpinee'

supplementary data
'**no ionization in ESI positive and negative mode.'
-> Consider evaporation of cpds before ionization instead or 'no ionization'. The problem is 'universal' ESIMS method
see, e.g., doi:10.3390/pharmaceutics12010007 (apart from their crazy-sure HRMS interpretations).

tabS3
-> Commas instead of points as separators. Correct.

== LANGUAGE ==

Readable and concise. Minor corrections needed, if any.

Author Response

Response to Reviewer 2 Comments

Point 1:

l.27, tables 1 and 3, text
MIC of the ethanolic extract is 4x 'better' than of essential oil.
-> It's possible that non-volatile phenolics are effective inhibitors of different enzymes (including also efflux pumps)
but
-> How to explain the high values of MIC for EO?
-> Can you consider insolubility/instability of solution/suspension during the assay? E.g., lack of stable mixing with assay broth? Evaporation of essential oil? Is the broth microdilution method in its unmodified version really applicable to essential oils? Can we compare soluble and non-soluble cpds with the same assay? Those are well-known problems.
-> Refer to the literature of this area. Discuss.

We know that p-cymene and g-terpinene posses other water-solubility than phenols. The situation can be, in part, similar to essential oil.
-> Please, refer to this problem in Discussion.

Response 1:

Thank you for your observations. Of course, the difference in MICs of the extract and essential oil could arise from differences in solubility and other technical challenges. To minimize the differences between tested preparations, all tested compounds/extracts were dissolved in the same solvent, DMSO. No precipitation or phase separation was observed when mixed with broth afterwards. Thus, we concluded that broth microdilution is appropriate for testing against C. jejuni in the given settings. The incubation of C. jejuni is carried out in anaerobic jars/boxes with microaerobic atmosphere. This makes excessive evaporation of a volatile and carryover within one plate more apparent, as it would interfere with growth controls. When all growth controls are as they should be, the results are considered valid. Because of the high MIC values, the essential oil, p-cymen and γ-terpinene MICs were confirmed by tube macro-dilution method. These results did match the microdilution results. We and other colleagues working with C. jejuni do consider the testing of essential oils dissolved in DMSO by the means of microdilution for antimicrobial activity against C. jejuni an appropriate and comparable method, providing all appropriate controls are used. Nevertheless, additional text explaining the method and results has been added to the method section in L142-L150, and discussion L400-L402, L421-L423, and L433-L437.

Point 2:

The essential oil has a much lower value of MIC (table 1, table 3).
-> Why, if its constituents work at least not-antagonistic (as a result of the other experiment, table 2)? Carvacrol and g-terpinene are 63+19 R% of essential oil.
-> Discussion and interpretation needed.

Response 2:

As not all constituents of the essential oil or ethanolic extract were tested we cannot be sure that no antagonism between compounds in these extracts would occur. Also, only binary combinations were tested in this study. Thus we do not feel entitled to comment on this matter without clearer evidence that would confirm additional claims. For some additional explanation, text has been added to the discussion section in L421-L423, and L429-L437.

Point 3:
-> What do we already know about the analyzed cpds AND closely related plants (Lamiaceae, THQ, THL, CRL, gTP, pCY, RA) in the aspect of actions reported in this work?

Response 3:

Additional text has been added in the discussion section L449-L451 to address this question.

Point 4:

Intro, l.74
The journal is devoted to 'foods.'
-> The hermetic abbreviations should be explained here or at the end list of abbrev.
-> Efflux pump types should be briefly described in Intro.

Response 4:

In the introduction section additional text has been added in L-53-L60, for better description of C. jejuni efflux pumps.

Point 5:
l.78
-> The reason to use 96% ethanol as extrahent should be explained.
-> This procedure does not guarantee high oil content; was the level of oil determined in the extract?

Response 5:

96% Ethanol was selected for two reasons, on the one hand it extracts compounds from a wide polarity range, on the other hand this solvent is also applicable in food processing. The oil content in the extract was not determined as the solvent had been evaporated under vacuum and additionally in a stream of nitrogen to remove remaining volatiles. Also due to sensory evaluation of the dried extract showed that easily volatile compounds were not present any more. HPLC analysis of the dried ethanolic extract revealed that only the phenolic carvacrol and or the quinone thymoquinone were present in higher amounts which is in agreement with their lower volatility. Nevertheless we analysed also the essential oil of the plant material and included p-cymene and γ-terpinene in further assays.

We changed the text as follows: Although no easily volatile compounds were detected in the ethanolic extract it became apparent that less volatile compounds of the essential oil like carvacrol and thymoquinone were also still present. Hence, in addition the composition of the essential oil was analysed by GC-MS and GC-FID (Supplementary Table S3). According to these analyses, carvacrol, thymol, thymoquinone, p-cymene, γ-terpinene, and rosmarinic acid were selected as the individual and combined pure test compounds.

For clarification, text has been added in L85-L87 and L227-L230.

Point 6:

l.88
-> To be clear, explain if the acetonitrile was also acidified.

Response 6:

The acetonitrile was not acidified, only the aqueous part of the mobile phase contained formic acid. Changes have been made to text in L98 to clarify.

Point 7:

fig1 caption, tabS2
-> 'Isomer' is unnecessary in all cases.

Response 7:

Corrected in figure 1 caption as well as in supplementary table S2.

Point 8:
-> If no standards for ms2 available, how can you know that it is rhamnoside but hexosyl. Shouldn't you name it deoxyhexoside for provisional identification, if you don't have a standard?
-> Correct.

Response 8:

Correct. We agree with the suggestion of the reviewer and changed the text L237 accordingly.

Point 9:

Materials and Methods
-> The purity of standards must be given (with briefly mentioned assay, e.g., GC-MS [producer declaration / internal protocol(reference)], HPLC-UV (program, lambda=...nm) or qNMR (standard, solvent, analytic shift position)).

Response 9:

All standards were commercial standards (Sigma) with specified purity. The purities have been added in text L121-L124.

Point 10:
-> Preparation and stability of stocks of water-not-well-soluble substances (THQ, THM, CRL...) should be given.

Response 10:

Stock of tested substances were always freshly prepared, before any tests, in DMSO. The text has been changed for better clarification in L142-L150.

Point 11:

l.26, 312, elsewhere
'y-terpinene' use special character 'gamma' instead 'y'; remember about italics

Response 11:

The character γ has been added instead of y before -terpinene.

Point 12:

l.131, 139, elsewhere
'ρ-'cymene -> 'p-'cymene
use italics for 'p-', everywhere

Response 12:

The conversion of 'ρ-'cymene to 'p-'cymene has been made.

Point 13:

decide to regularly use or not to use caps after a hyphen, in chemical names

Response 13:

Thank you for the observation. We decided to remove caps.

Point 14:

use non-breakable space between C. and jejuni, everywhere

Response 14:

All C. jejuni have non-breakable spaces now throughout the text.

Point 15:

l.261
last line of tab2, correct 'terpinee'

Response 15:

The error has been corrected in table 2.

Point 16:

supplementary data
'**no ionization in ESI positive and negative mode.'
-> Consider evaporation of cpds before ionization instead or 'no ionization'. The problem is 'universal' ESIMS method
see, e.g., doi:10.3390/pharmaceutics12010007 (apart from their crazy-sure HRMS interpretations).

Response 16:

This is a good point raised by the reviewer. The possibility of evaporation of the compounds during ESI cannot be excluded and hence we changed the text accordingly to “no signal in ESI positive and negative mode”.

Point 17:

tabS3
-> Commas instead of points as separators. Correct.

Response 17:

Commas have been changed into points in supplementary table 3.

Reviewer 3 Report

Peer review of Šimunović et al, The common culinary herb winter savory against the infamous food pathogen Campylobacter jejuni

 This manuscript uses classic microbiology approaches to test the minimum inhibitory concentration of extracts and pure compounds from the winter savory herb against a strain of the food pathogen Campylobacter jejuni. The pure compounds were selected based on their abundance in the ethanolic or essential oil extract. In addition to studying the pure compounds in isolation, synergistic effects were assessed on combinations of these compounds. Finally, the authors use deletion mutants of C. jejuni to study the role of efflux pump genes in the sensitivity against the pure compounds and the extracts to find that the complex extracts.

Overall the work seems to be of solid quality. However, I feel that the authors have to make the following improvements

My concerns:

  1. Please spell out in the discussion where the study adds novelty. From the current discussion it seems that most if not all of the findings have already been reported previously.
  2. In Figure 2C and 3C statistics (e.g. ANOVA) have to be added.
  3. Figure 2C x-axes labels has a typo; “p-cymen”. Y-axis should be fold-change, not fold increase
  4. I found the paper easy to follow until Table 2. The authors should clarify / discuss in text why Amix and Bmix values are different. The way I understand it is that both are a combination of the same two compounds so why can there be 2 mixes / concentrations? It could very well be that this is my mistake in misunderstanding, but that is a sign that the authors did not clarify what they presented here.
  5. Please clarify the principle of the EtBr assay. Is the EtBr only fluorescent when it is in the cell or are the cells washed before measuring?
  6. In the methods please clarify that resazurin is used to assess cellular respiration as a measure of bacterial growth. It is thus not a direct measure of MIC as is suggested by te wording in the methods.
  7. Please add a discussion on the possible mechanisms of the synergism of the compounds, for instance carvacrol and thymol
  8. Last sentence of the results is not warranted unless expression of CmeABC is actually verified in this specific CmeR deletion strain
  9. In the conclusion rather than say “further testing is needed” specify what kind of testing is needed.

Author Response

Response to Reviewer 3 Comments

Point 1:

Please spell out in the discussion where the study adds novelty. From the current discussion it seems that most if not all of the findings have already been reported previously.

Response 1:

Additional text has been added in the discussion to clarify this matter. L381-L385 and L486.

Point 2:

In Figure 2C and 3C statistics (e.g. ANOVA) have to be added.

Response 2:

Statistics has been added to the figures.

Point 3:

Figure 2C x-axes labels has a typo; “p-cymen”. Y-axis should be fold-change, not fold increase

Response 3:

Thank you for the observation. The error has been corrected in Figure 2.

Point 4:

I found the paper easy to follow until Table 2. The authors should clarify / discuss in text why Amix and Bmix values are different. The way I understand it is that both are a combination of the same two compounds so why can there be 2 mixes / concentrations? It could very well be that this is my mistake in misunderstanding, but that is a sign that the authors did not clarify what they presented here.

Response 4:

Additional text was added for explanation of each concentration in L283-L286.

Point 5:

Please clarify the principle of the EtBr assay. Is the EtBr only fluorescent when it is in the cell or are the cells washed before measuring?

Response 5:

EtBr is an intercalating agent that emits a fluorescent signal. Within the cell it binds to DNA and the increased fluorescence is a sign of EtBr accumulation in the cell. Cells are washed prior to EtBr addition, as it is stated in the materials and methods section. We have added additional explanations to the method (L189-L1190) and results (L292-L293) sections.

Point 6:

In the methods please clarify that resazurin is used to assess cellular respiration as a measure of bacterial growth. It is thus not a direct measure of MIC as is suggested by te wording in the methods.

Response 6:

The resazurin MIC detection method has been extensively tested in our and other laboratories working with C. jejuni, and has shown good detection limit of >5x105 CFU/mL and reproducibility. Additional text has been added to the methods section for clarification L150-L153.

Point 7:

Please add a discussion on the possible mechanisms of the synergism of the compounds, for instance carvacrol and thymol

Response 7:

Additional text has been added in the discussion to attend this question, in L429-L432 and L449-451.

Point 8:

Last sentence of the results is not warranted unless expression of CmeABC is actually verified in this specific CmeR deletion strain

Response 8:

The expression of CmeABC was not tested in this articular mutant, thus the sentence was removed. L375-L377.

Point 9:

In the conclusion rather than say “further testing is needed” specify what kind of testing is needed.

Response 9:

Thank you for this suggestion. Additional text has been added to the conclusion in L497-L498.

Round 2

Reviewer 2 Report

Thank you.

The paper is clear enough for me, now.